# How Close Are We to Patient-Side Troponin Testing?

**DOI:** 10.3390/jcm13247570

**Published:** 2024-12-12

**Authors:** Aaron Goldberg, Samuel McGrath, Michael Marber

**Affiliations:** 1Stoke Mandeville Hospital, Aylesbury HP21 8AL, UK; aaron.goldberg1@nhs.net; 2BHF Centre of Research Excellence, The Rayne Institute, King’s College London, 4th Floor, Lambeth Wing, St Thomas’ Hospital, London SE1 7EH, UK

**Keywords:** point of care (POC), high-sensitivity cardiac troponin (hs-cTn), acute coronary syndrome (ACS)

## Abstract

Laboratory-based high-sensitivity cardiac troponin testing has been the pillar for emergency stratification of suspected acute coronary syndrome for well over a decade. Point-of-care troponin assays achieving the requisite analytical sensitivity have recently been developed and could accelerate such assessment. This review summarises the latest assays and describes their potential diverse clinical utility in the emergency department, community healthcare, pre-hospital, and other hospital settings. It outlines the current clinical data but also highlights the evidence gap, particularly the need for clinical trials using whole blood, that must be addressed for safe and successful implementation of point-of-care troponin analysis into daily practice. Additionally, how point-of-care troponin testing can be coupled with advances in biosensor technology, cardiovascular screening, and triage algorithms is discussed.

## 1. Introduction

Acute coronary syndrome (ACS) has a major impact on patients, healthcare services, and clinicians. Ischaemic heart disease causes nine million annual deaths globally [1], and suspected ACS accounts for 10% of emergency department (ED) visits [2]. Timely, accurate, and safe triage of such patients is thus imperative. Since their advent in the 1980s [3], immunoassays for cardiac troponin (cTn) as myocardial injury indicators have played a central and increasingly dominant role in ACS assessment. Cardiomyocytes are the sole source of cTnI and the main source of cTnT; hence, these assays have high biological specificity and are the only recommended biomarker for acute myocardial infarction (AMI) diagnosis in the Fourth Universal Definition of Myocardial Infarction [4]. Central laboratory testing (CLT) has been the mainstay of this cTn measurement. However, CLT’s drawbacks include result turnaround times of up to one hour [5], the need for trained laboratory personnel, and accessibility issues in resource-poor healthcare settings. Promisingly, point-of-care (POC) analysers seem adept to address these challenges.

Older cTn assays had optimum sensitivity 10–12 h after AMI onset, resulting in patients needing admission for observation and serial troponin measurement. High-sensitivity cardiac troponin (hs-cTn) measurement by CLT was developed in 2009 [6]. These platforms can detect lower troponin concentrations occurring earlier in AMI, and so have been the focus of ACS biochemical diagnosis. These hs-cTn assays are defined by two key analytical attributes [7]. Firstly, their minimal variation at the cut-off cTn value for diagnosing AMI amongst a healthy reference cohort (coefficient of variation (CV) at 99th percentile upper reference limit is ≤10%). Secondly, they are sensitive enough to detect very low cTn levels in more than half of healthy individuals (measurable concentrations above limit of detection (LoD) for >50% healthy population). Trials have evidenced CLT-based hs-cTn measurements’ ability to rule in AMI with a single high cTn level, rule out AMI with a single low result, and then triage those with intermediate levels using the change in concentration on repeat measurements [8]. Accordingly, international guidelines, such as the European Society of Cardiology (ESC), recommend the use of such testing in rapid algorithms for patients presenting with suspected non-ST-segment elevation myocardial infarction (NSTEMI) [9]. 

POC cTn testing can be performed faster and closer to the patient than CLT, presenting an exciting potential solution for accelerating suspected NSTEMI work-up. The first contemporary POC assays were rightfully usurped by the superior sensitivity of hs-cTn CLT, but POC instruments with improved analytical abilities have recently been developed. This review aims to provide non-specialist clinicians with an overview of how these new high-sensitivity POC troponin tests may benefit the chest pain management armamentarium. We discuss what is already known about their clinical performance and what key clinical information is still needed for their safe and effective implementation into daily practice.

## 2. What Is Meant by POC

POC testing is a diagnostic measurement performable at or near the patient by non-laboratory staff with swift results, potentially influencing the care provided. It needs to be operator-friendly and capable of utilising whole-blood samples. This negates the need for preparatory steps common in CLT such as pipetting and centrifugation which are required to process serum or plasma samples. Compared to the longer turnaround times of conventional CLT, POC analysis can yield results in 5–15 min [10]. POC testing has been utilised to assess a range of biochemical parameters, such as electrolytes, full blood counts, and infection markers. This has been shown to generate faster management and disposition decisions that can reduce the length of patients’ ED stay [11].

Translating these benefits into suspected ACS assessment is very attractive. The rationale for POC cTn testing is to directly give results faster to the responsible clinician and facilitate testing in a wider range of healthcare settings. A key sequela would hopefully be quicker clinical decision-making that expedites patients to the correct care pathway, enabling faster discharge of low-risk patients to alleviate crowded EDs. It could also allow for quicker provision of therapeutic intervention when needed for confirmed NSTEMI and prevent unindicated ED visits altogether if utilised effectively in the community. POC cTn instruments vary in scale from smaller portable battery-powered devices to desktop machines which can be situated within the clinical workspace but may need a wired power supply [12]. They are immunoassays, using labelled antibodies to detect cTnI or cTnT, and some can also run simultaneous assays for other biochemical markers. Table 1 summarises the specifications of the three current hs-cTn POC devices.

## 3. The Concentration Problem

POC cTn assays are ideally suited for rapidly ruling in AMI diagnosis. It has been demonstrated that POC can reliably rule in MI by detecting 99th percentile upper reference limit troponin levels with a high predictive value [14,15]. However, safely and effectively ruling out AMI is the crux of managing patients presenting with ACS symptoms. In the context of a normal or unchanged ECG, this necessitates either an undetectably low troponin level close to or below the assay’s LoD or a measurably low level which shows unchanging repeat measurements. The LoD is defined as the lowest analyte concentration reliably distinguishable from a blank sample and at which detection is feasible [16]. These requirements are agreed by international guidelines, albeit with variation in the advised timing for serial measurement (e.g., 0/1 h or 0/3 h to rule out) [9,17]. Laboratory hs-cTn assays are the current gold standard for accomplishing this, setting the benchmark for POC tests. Achieving this degree of clinical sensitivity presents POC assays, using a single approximately 100-microlitre droplet of blood, with an analytical concentration problem. POC analysers measure concentrations of cTnI, which has an AMI rule-out concentration of 2 ng/L [18]. This equates to an incredibly small 10–100 femtomolar (10–100 × 10^−15^ molar) concentration in the POC sample volume. Therefore, to detect this very low level of cTnI, POC assays must surmount the challenge of achieving the requisite high sensitivity with just a drop of blood.

Previously available POC assays could not perform such high-sensitivity analyses necessary to rule out MI on a single admission blood test. They may even have had a diagnostic performance worse than a triage decision aid alone [19]. Instead, these contemporary assays required serial sampling at three to six hours post-presentation to attain sufficient rule-out sensitivity [20]. Therefore, CLT is clearly preferable to exclude or diagnose NSTEMI within one to three hours of admission compared to the extra time requirements of a POC instrument’s repeat measurements. The advantage of a 10–15 min turnaround time of a conventional POC cTn assay is swamped by the delay needed to match the diagnostic sensitivity of CLT hs-cTn assays. However, as per the IFCC C-CB [13], there are now currently three POC machines meeting hs-cTn assay analytical criteria, namely Atellica VTLi (Siemens Helathineers, Erlangen, Germany), PATHFAST (LSI Medicine Corporation, Tokyo, Japan), and TriageTrue (QuidelOrtho, San Diego, CA, USA); their assay characteristics are outlined in Table 1.

Studies of Atellica VTLi’s diagnostic performance indicate that it is comparable with hs-cTn CLT and suggest that it could be utilised in rapid rule-out algorithms [21,22,23]. Similar studies have been performed for PATHFAST [24] and TriageTrue [25]. These initial analyses do unfortunately have a significant drawback; they were performed on biobanked or processed plasma samples under controlled laboratory conditions. Therefore, such results have dubious applicability to the ED setting, in which non-laboratory clinicians would use whole-blood samples for POC analysis without preparatory steps. The National Institute for Health and Care Excellence similarly emphasises the importance of assessing POC’s diagnostic performance with whole-blood samples [26]. However, there are very recent studies attempting to bridge this applicability gap to real-world practice. Compared to the CLT hs-cTn gold standard, Atellica VTLi has shown a strong correlation using whole-blood samples over a wide range of cTnI concentrations [27,28,29] and concordant NSTEMI diagnostic performance for non-traumatic chest pain in the ED [30]. Furthermore, it has shown good equivalence with CLT for accurate MI diagnosis within a 0/2 h serial testing protocol of whole blood from patients presenting to the ED with suspected cardiac ischaemia [31,32]. Similar whole-blood analytical correlation with a CLT benchmark has also been demonstrated for PATHFAST [33]. Furthermore, there are other POC assays that are close to achieving the required degree of clinical sensitivity [34,35,36]. However, there is a need to bridge the evidence gap pertaining to whether the equivalence in analytic performance between POC testing and CLT translates to better economic or health outcomes. This requires prospective clinical trials where the two systems drive clinical decisions.

## 4. Potential Advantages of POC Testing

### 4.1. Ambulance Triage

The portability of POC cTn assays means that there is certainly scope for their use in ambulance triage of acute chest pain cases. The collection of cTn levels with concomitant ECG analysis by paramedics is feasible [37]. This prevents the need for secondary inter-hospital transfer as high-risk NSTEMI patients are identified earlier and transported initially to centres with percutaneous coronary intervention facilities. Recent trials demonstrate that in comparison to ED transfer, the utilisation of ambulance POC cTn assays to identify patients as low-risk for ACS reduces incurred healthcare costs without increasing major adverse cardiovascular event (MACE) outcomes at 30 days [38,39,40]. Furthermore, studies have conveyed that incorporating POC cTn results into the pre-hospital History, ECG, Age, Risk, and Troponin (preHEART) score performs better than other triage scores without POC results and POC cTn testing alone [41,42]. This enables a significant proportion of low-risk patients to avoid hospitalisation [43]. However, there is conflicting evidence that whilst pre-hospital POC cTn shows high specificity, it lacks sensitivity compared to CLT, and hence cannot safely rule out MI [44]. Ambulance crews attend to patients sooner from symptom onset than when venepuncture is performed in the ED for CLT; thus, the low sensitivity in these studies may arise from the troponin-blind period during ACS progression. Additionally, the ability to rule in AMI alone would still help guide pre-hospital management, such as antiplatelet administration, transport urgency decisions, and conveying patients directly to cardiac catheterisation centres. The above studies all used either Cobas h232 (Roche Instr, Basel, Switzerland) or i-STAT (Abbott Point of Care, Princeton, NJ, USA) POC assays, which, whilst they can be handheld, do not meet hs-cTn analytical standards. Therefore, pre-hospital studies assessing the utility and safety of hs-cTn POC assays small enough to be deployed in ambulances, such as Atellica VTLi and TriageTrue, are needed.

### 4.2. Primary Care and Other Community Healthcare Settings

There are other community settings whereby POC cTn testing could be advantageous. Primary care physicians commonly encounter patients presenting with chest pain [45]. Amongst these patients, those at a low risk of ACS could avoid unnecessary urgent hospital transfer via risk stratification incorporating POC cTn in primary care, provided other severe acute differential diagnoses are not clinically suspected. An accelerated diagnostic chest pain pathway in general practice using POC cTn identified over 60% of patients as low-risk who were resultingly managed in the community safely without 30-day MACE [46]. POC testing could prove similarly beneficial for rural or resource-poor healthcare settings with more limited access to CLT. In rural Australia, a cardiac support model, in which tertiary centre specialists remotely reviewed ECGs and POC cTn results to advise on management and onwards referral, reduced missed AMI diagnoses and improved rates of primary reperfusion therapy and 30-day mortality [47,48]. A similar chest pain management pathway involving POC cTn testing demonstrated safe and effective identification of low-risk ACS patients in rural New Zealand too [49]. However, these benefits need to be considered in country-specific healthcare contexts and do not necessarily translate into lower-income countries. For example, the HEART triage score including POC cTn could not safely identify low-risk patients in Tanzania. This was possibly due to patients presenting later, worse access to coronary angiography, and lesser uptake of evidence-based treatment and secondary prevention for ACS [50]. Figure 1 summarises the various community healthcare settings in which POC cTn testing may prove valuable.

### 4.3. ED

We must first consider the current landscape of suspected ACS work-up in the ED to appreciate how POC cTn testing could be a beneficial addition. There is clear and strong evidence that rapid rule-out pathways incorporating hs-cTn assays, which can be on a 0/1 [51], 0/2 [9], or 0/3 h basis [52], safely accelerate triage without harming patient outcomes, hence their recommendation in international guidelines. Such protocols can shorten the length of stay for patients in the ED and increase the rate of direct ED discharge when AMI is ruled out, without excess diagnostic resource utilisation [53]. Moreover, clinical scoring systems containing hs-cTn values, such as HEART [54] and Emergency Department Assessment of Chest Pain Score (EDACS) [55] pathways, have demonstrated consistent reductions of 20–45% in suspected ACS admissions. These decreased admission rates, combined with shorter subsequent inpatient stays, translate to significant cost savings for hospitals [56]. Additionally, economic benefits are further enhanced by reduced rates of downstream diagnostic procedures, such as functional testing [57] and anatomic imaging including invasive coronary angiography [58]. This swifter assessment does not compromise patient outcomes, with multicentre studies illustrating that using hs-cTn strategies to triage patients as low-risk for ACS does not impact the incidence of mortality or MI at 30 days [18,59,60]. The clinical reliability, safety, and benefits of hs-cTn algorithms have additionally been affirmed via meta-analysis [8,61]. 

Despite this clear evidence of benefits, such protocols are not universally deployed day-to-day in EDs. A survey of 1902 medical centres across five continents in 2019 revealed that only 41% utilise hs-cTn assays and less than 10% implement 0/1 or 0/2 h algorithms [62]. In fact, when accelerated protocols are indeed implemented, significant intracentre variation in ED stay duration has been highlighted [53]. A likely explanation for this is the infrastructural challenge of CLT hs-cTn, including delays portering blood samples to a centralised lab from busy EDs, sample processing, available access to sensitive analysers, and clinical teams being aware of the results (push versus in-record notification). POC instruments, with their short result turnaround times of under 20 min [31] and ability to be located within the ED itself, appear ideally suited to circumvent such barriers and help expedite patient flow.

Older POC tests, which as aforementioned require serial cTn testing, have yielded mixed findings regarding their benefits in the ED. Contrasting inter-hospital results arose for successful discharge and cost savings from using a panel POC assay including troponin [63], and POC cTn testing only minimally reduced the time from first medical contact to ED discharge by 0.3 h [64]. However, more recent studies indicate that incorporating POC cTn testing, including assays meeting hs-cTn criteria, into the ED triage process is safe, accurate, and timely. The inclusion of POC results in the troponin-only Manchester ACS decision aid may facilitate ruling out suspected ACS within three hours for nearly one-third of ED patients without missing any AMIs [65]. A prospective observational study of two large ED cohorts has derived and validated a 0/2-hour strategy using Atellica VTLi, enabling earlier clinical decision-making without augmenting 30-day MACE [66]. Figure 1 portrays how POC cTn assays could be beneficial within EDs in this way. POC testing may be especially adept for supporting self-contained dedicated chest pain units, which have been previously shown to decrease hospital admissions and costs and improve patient outcomes and satisfaction [67,68].

There are caveats to ED POC cTn testing that would need to be carefully considered for widespread implementation. Firstly, indiscriminate troponin testing is a potential by-product of the increased accessibility afforded by POC testing in the ED. This can lead to uncertain diagnosis and further unneeded investigations. A sizeable trial highlighted this; the implementation of an hs-cTnI lab assay reclassified a significant proportion of suspected ACS patients as higher risk, but only a third of these did indeed have type 1 MI and there was no improvement in adverse cardiovascular outcomes [69]. Secondly, the potential advantages need to be regarded in the context of other hurdles to POC implementation in the ED. Notably, these include the costs of establishing POC analysers in situ and maintaining them, the analytical errors arising from operation by clinicians not trained in laboratory medicine, and the increased workload for busy ED staff [70,71]. These factors are likely to be prominent during initial implementation and would improve over time as the ED staff’s familiarity with the analysers grows. Thirdly, cTn analysis is not inherently the rate-limiting step in chest pain assessment because appropriate work-up for other differentials, such as performing d-dimer, chest X-ray, and pulmonary angiogram for pulmonary embolism, is crucial. Additional investigation results, such as haemoglobin, electrolytes, and renal function, are also imperative in triaging ACS patients. Thus, POC testing’s faster result times will not translate into more efficient clinical practice unless it is well integrated into wider patient management systems.

### 4.4. Elsewhere in Hospitals

POC cTn testing may also have advantageous roles in hospitals outside of the ED ACS triage setting. The ESC guidelines recommend perioperative troponin measurement for patients undergoing intermediate- or high-risk surgery because myocardial injury in this setting can increase long-term morbidity and mortality risk [72]. Troponin levels are also a suggested component of monitoring during cardiotoxic cancer therapy [73]. Therefore, POC cTn analysers may benefit specialist surgical or oncological centres with more limited access to a central laboratory. Additionally, a POC instrument could serve as an efficient back-up analyser in the case of hs-cTn CLT malfunction or when lab-trained staff are not available overnight. It must be emphasised that different cTn assays are not harmonised and have non-standardised individual cut-offs; thus, lab and POC results should not be directly compared [74] and need to be clearly flagged in the patient’s record.

## 5. Trial Experience

There is clearly evidence from observational studies, as described above, that there are now POC assays reaching hs-cTn analytical standards, and that utilising POC cTn testing in pre-hospital and ED environments could be safe and effective for ACS triage. However, these findings are limited by observational studies’ increased bias and inability to prove causality [75]. RCTs comparing POC testing with standard CLT in clinical practice can reveal how the former’s rapid result turnaround time directly impacts clinical decision-making and patient outcomes; the handful of such trials currently published are summarised in Table 2. Crucially, they were all performed on pre-hs-cTn POC assays. The findings are variable, particularly regarding POC testing’s impact on shortening the decision time from admission to disposition for suspected ACS patients. Within multicentre RCTs, disparate effects on length of stay were found between EDs [63,76]. The lack of standardised chest pain protocols between hospitals, such as different assessment time targets, could explain the inconsistent trial findings. This highlights that POC cTn testing implementation needs to be well integrated within the wider ED triage pathway [71]. Understandably, these RCTs were designed pragmatically, and thus it was not possible to blind clinicians to which assay produced the provided cTn result. This may have attenuated POC testing’s actual impact as irrespective of a swift POC result, clinicians might have waited for the validating CLT value [77] or gone on to order further lab tests anyway. Regardless of these limitations, the RCTs indicate significant scope for POC cTn testing to accelerate suspected ACS assessment, and none identified any major risks for patient safety in doing so.

## 6. What the Future May Bring

Going forwards, there is a clear need for RCTs that compare hs-cTn testing with POC and laboratory assays in the clinical ED setting using unspun whole-blood samples. The ICare-FASTER trial is such a study of the Atellica VTLi analyser; it has been protocolised [82] and will hopefully provide invaluable information on POC hs-cTn’s actual clinical impact on ED length of stay, morbidity, and mortality. The POB HELP RCT is also underway to assess how using the Atellica VTLi assay with fingerstick samples impacts patient flow and ACS rule-out in primary care [83]. Factors specific to POC instruments will need to be borne in mind in such studies. For example, pre-analytical errors can arise from haemolysis, which is more common with unspun whole blood [84], and the relationship between venous and fingerstick capillary cTn levels needs to be better understood [12]. 

In the future, POC cTn measurement may be deployed for detecting chronic cardiovascular disease risk in asymptomatic populations. It has been well demonstrated that cTn is a strong predictor for cardiovascular morbidity and mortality; cTnT has been suggested as a better predictor for all-cause mortality [85] and cTnI for future AMI [86]. There are no consensus-agreed preventative interventions once such increased risk has been identified, and thus routine troponin screening is not recommended. However, should screening and specific interventions be shown to lower chronic cardiovascular risk, POC cTn analysers in the community would be ideal for accomplishing this. Detecting early cTn elevations in this way could reveal subclinical cardiac disease and therefore provide an opportunity to deliver targeted preventative therapy [87]. 

Technological advances could make POC cTn testing even more accessible and easy-to-operate. In comparison to more traditional biomarker methods like enzyme-linked immunosorbent assays, considerable progress has been made in the field of microfluidic biosensors. These have the potential to usher in more cost-effective, miniaturised, and ergonomic detection systems [88]. For example, a wrist-worn transdermal infrared spectrophotometric sensor has been shown to accurately detect elevated cTnI levels amongst inpatients with confirmed ACS [89]. Additionally, there is mixed evidence that saliva can provide samples for rapid and simple troponin analysis [90]. These technologies could enable easy POC cTn measurements in a broader range of settings, including possible “plug-and-play” style and smartphone-compatible systems [91,92]. However, they all certainly still need further validation studies and regulatory clearance. As mentioned earlier, POC cTn can be beneficially incorporated into ACS triage scores, but given machine learning progress, they can also be implemented into more complex stratification tools. Compared to international guideline-recommended pathways, a recent artificial intelligence algorithm integrated single POC hs-cTn measurements alongside other clinical variables, such as ischaemic ECG changes, smoking status, and family history of ACS, to identify more patients suitable for ACS rule-out while maintaining high safety [93]. Going forwards, these algorithms should not be used in isolation to guide management but rather should help guide shared decision-making with patients.

## 7. Conclusions

POC instruments meeting hs-cTn assay criteria are now available. This makes them viable alternatives to the CLT recommended in international ACS guidelines, with their added benefits of faster result turnaround times, operability closer to the patient, and user-friendly form factor. Therefore, potential avenues for POC cTn testing to benefit AMI diagnosis in community and ED settings are evident. They are also well suited for integration with advances in biosensor, artificial intelligence, and screening tools. However, prospective device evaluation using whole blood in collaborative RCTs between manufacturers and researchers is still required for POC testing’s future safe clinical deployment. Hopefully, these trials will demonstrate that POC hs-cTn testing can safely and effectively accelerate ACS triage, leading to improved patient outcomes and galvanising further progress in this field.

## Figures and Tables

**Figure 1 jcm-13-07570-f001:**
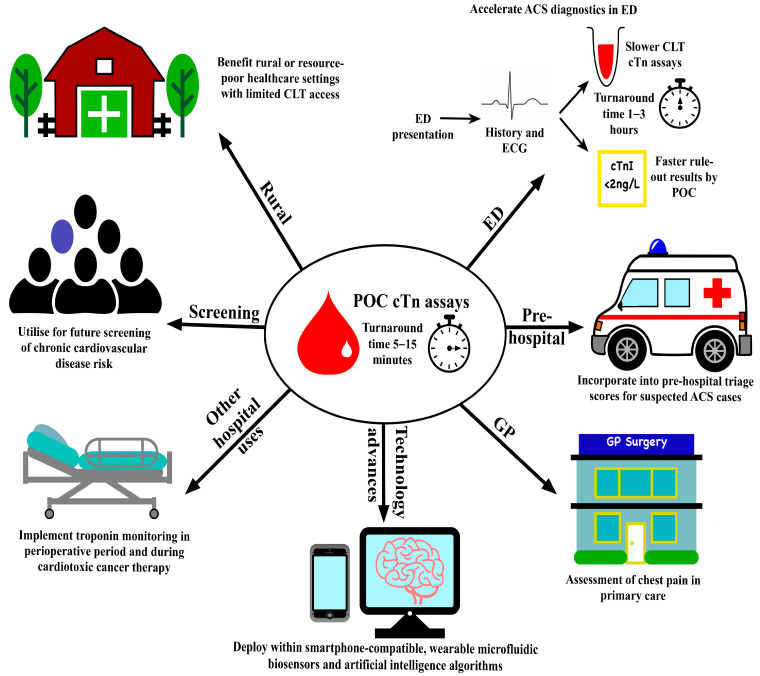
Summary of POC cTn assays’ varied potential beneficial healthcare applications going forwards.

**Table 1 jcm-13-07570-t001:** Specifications and analytical characteristics of current hs-cTn POC analysers as per International Federation of Clinical Chemistry Committee of Clinical Application of Cardiac Bio-Markers (IFCC C-CB) [13].

Analyser	Instrument Size (Width × Length × Height in cm) *	Assay	Sample Type	Time to Results (Minutes)	LoD (ng/L)	% Measured > LoD in Healthy Population	99th Percentile (ng/L)	%CV at 99th Percentile (Under Optimised Lab Conditions)
Atellica VTLi (Siemens Healthineers, Erlangen, Germany)	Portable desktop system; 8.5 × 25 × 5.2	hs-TnI	Heparin-Li whole blood, plasma, or capillary blood	≈8	1.2 (plasma), 1.6 (whole blood)	Overall 84, females 80, males 87	Overall 22.9, females 18.5, males 27.1	6.5 (plasma), 6.1 (whole blood)
PATHFAST (LSI Medicine Corporation, Tokyo, Japan)	Compact desktop system; 34 × 57 × 48	hs-TnI	Heparin-Na, heparin-Li, or EDTA whole blood or plasma	<17	2.9	Overall 66.3, females 52.8, males 78.8	Overall 27.9, females 20.3, males 29.7	6.1
TriageTrue (QuidelOrtho, San Diego, CA, USA)	Portable bedside system; no dimensions provided	hs-TnI	EDTA whole blood or plasma	<20	0.7–1.6 (plasma), 1.5–1.9 (whole blood)	Overall ≥50	Overall 20.5, females 14.4, males 25.7	5.0–5.9 (plasma), 5.9–6.5 (whole blood)

* Where available, analysers’ dimensions taken from respective manufacturers’ websites. hs-TnI, high-sensitivity troponin I; LoD, limit of detection; CV, coefficient of variation.

**Table 2 jcm-13-07570-t002:** Published RCTs comparing POC testing and CLT of cTn in clinical practice.

Trial	cTn Assays Compared (POC vs. CLT) *	MI Diagnosis Protocol	Outcomes Measured	Results	Key Limitations
Multicentre RCT of EDs in USA [76]	Abbott i-Stat cTnI (Abbott Point of Care, Princeton, NJ, USA) vs. unspecified CLT cTnI assay	Serial cTn testing over 6, 8, or 12 h; no cut-off value specified	Time from admission to discharge or transfer to inpatient cardiology ward	Inconsistent changes in length of ED stay between EDs	Sampling bias introduced by exclusion of large proportion of patients after randomisation
Single-centre RCT in Australian ED [77]	AQT Flex cTnT (Radiometer, Copenhagen, Denmark) vs. Roche hs-cTnT	At least one 0/6-h cTn value > 14 ng/L	MACE at 3 months	No significant difference	Small sample size and unblinded clinicians
RATPAC multicentre RCT in UK EDs [78]	Siemens Stratus CS panel (Dade Behring, Milton Keynes, UK) vs. Siemens cTnI ultra, Abbott cTnI, Beckman Accu TnI, or Roche cTnT	Any cTn value > 0.07 μg/L or delta change between tests on admission and after 90 min	Successful ED discharge < 4 h, MACE at 3 months, and length of hospital stay	More early discharges, fewer admissions, and no significant difference in MACE	Unblinded clinicians
Single-centre RCT of coronary care unit admissions in UK [79]	Roche Cardiac T test (Roche Diagnostics, Lewes, UK) vs. Bayer Axon cTnT (Bayer Diagnostics, Newbury, UK)	cTn > 0.2 μg/L at 12 h after admission	Length of hospital stay and 6-month mortality	Shortened hospital stays with no significant difference in mortality	Small sample size
Two-centre cluster RCT in Australian EDs [80]	Abbott i-Stat cTnI (Abbott Point of Care, Princeton, NJ, USA) vs. Beckman Coulter Accu TnI assay (Beckman Coulter, Fullerton, CA, USA)	Not provided	Length of ED stay	Insignificant reduction in length of ED stay	Underpowered as intervention not mandated so significant proportion of patients did not receive POC testing
Single-centre RCT in ED in France [81]	Siemens Stratus CS cTnI (Stratus CS Test Systems, Dade Behring, Marburg, Germany) vs. Dimension RxL-HM cTnI (Dade Behring, Newark, DE, USA)	cTn > 0.1 μg/L but timing not provided	Time to NSTEMI therapy initiation and length of stay in ED	Reduced time to treatment commencement but no reduction in ED stay length	Small sample size and unblinded clinicians

* Manufacturer names and locations provided when exact POC or CLT cTn testing device specified in RCT. cTn, cardiac troponin; MACE, major adverse cardiovascular event; NSTEMI, non-ST elevated myocardial infarction.

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
