# Peer review of "How Close Are We to Patient-Side Troponin Testing?"

_jcm, 2024, doi:10.3390/jcm13247570_

Round 1

Reviewer 1 Report

Comments and Suggestions for Authors

The Authors submitted a good synthesis of the literature regarding the advantages and disadvantages of point-of-care troponin testing. The main ideas are presented fluently, in a logical sequence. Available evidence is critically analyzed. Written in an organized manner, this manuscript achieves the goal proposed by the Authors, namely to provide non-specialist clinicians an overview of how these new high sensitivity POC troponin tests may be useful in chest pain management.

Minor comments:

Lines 63-64: Compared to the longer turnaround times of conventional CLT…

To emphasize the significant amount of time saved by using POC assay instead of CLT assay, I recommend adding the time required to obtain the cTn value in CLT.

The size of TriageTrue is 21.6×15.8×6.4 cm and is provided by this article doi: 10.21037/jlpm-23-3

(The Authors are free to use this information or not, as it is not provided directly by the manufacturer)

Line 163: Comment:  TriageTrue and Atellica VTLi are portable devices, while PATHFAST is benchtop. I suggest the following change: … small enough to be deployed in ambulances, such as Atellica VTLi and TriageTrue, are needed.

Lines: 188 – 189. The use of 0h/2h algorithm is also supported by ESC guidelines and it should be mentioned along with the other two protocols. Furthermore, the 2023 ESC Guidelines for the management of acute coronary syndromes recommends to use the 0 h/1 h algorithm (best option) or the 0 h/2 h algorithm (second-best option).

Line 208: A likely aetiology… Cause/explanation is more suitable.  

Thank you!

Reviewer 2 Report

Comments and Suggestions for Authors

This paper reviews the advancements in point-of-care (POC) high-sensitivity cardiac troponin (hs-cTn) assays and their potential to revolutionize the diagnosis and management of acute coronary syndrome (ACS). It highlights the advantages of POC testing, including faster results and greater accessibility in emergency, pre-hospital, and community settings, compared to traditional central laboratory testing (CLT). The authors discuss the current limitations, such as integration challenges and the need for clinical validation using whole blood, while emphasizing the future potential of these devices to improve patient outcomes through innovative technologies like biosensors and artificial intelligence.

Comments and Suggestions for Authors:

  1. Abstract:
    • The abstract is clear and informative but could be more concise. Focus on summarizing the key findings, such as the clinical utility of POC hs-cTn devices, the gaps in evidence, and future research needs. Avoid repeating detailed background information that is covered in the introduction.
  2. Introduction:
    • The introduction effectively sets the stage by discussing the global burden of ACS and the importance of timely diagnosis. To enhance engagement, include a brief mention of the key challenges with traditional CLT and the promising role of POC testing early on. This will hook the reader by highlighting the relevance of the topic.
  3. Figures and Tables:
    • Ensure that all figures, such as Figure 1, are included and visually clear. Consider adding a flow diagram or graphical abstract summarizing the applications and benefits of POC hs-cTn testing across different settings.
    • Table 1 is comprehensive but could be visually enhanced. Bold key features such as Limit of Detection (LoD), sample type, and time to results to make comparisons more intuitive. A summary row highlighting the best-performing device for specific use cases could also be useful.
  4. Discussion:
    • The discussion provides a balanced view of the potential advantages and limitations of POC testing. However, consider elaborating on the following points:
      • Economic and Logistical Challenges: Discuss the cost of implementing POC devices, including maintenance and training requirements for non-laboratory staff.
      • Integration into Clinical Pathways: Highlight the need for integrating POC results into broader clinical workflows to ensure they contribute meaningfully to patient care. Provide examples of successful integration in specific settings, such as rural hospitals or emergency departments.
      • Risk of Overuse: Address the potential risk of overusing troponin testing due to the increased accessibility of POC devices and how this might lead to unnecessary investigations.
  5. Conclusion:
    • The conclusion summarizes the findings well but could be made more impactful by including specific calls to action. For example:
      • Encourage collaboration between device manufacturers and clinical researchers to address evidence gaps.
      • Highlight the importance of conducting randomized clinical trials (RCTs) using whole-blood samples in real-world settings to validate POC devices.
  6. Future Directions:
    • The mention of biosensors, artificial intelligence, and wearable technologies is forward-thinking. Consider discussing specific examples or ongoing projects to illustrate the potential of these innovations. For instance:
      • A brief description of how AI could enhance triage decisions by integrating POC test results with other clinical data.
      • Mention wearable troponin sensors or smartphone-compatible diagnostic tools as emerging technologies that could revolutionize POC testing.
  7. Practical Implications:
    • Add a brief section discussing how POC devices could be particularly impactful in underserved healthcare settings, such as low-income or rural areas. This will broaden the relevance of the study and appeal to a wider audience.
  8. Suggestions for Additional Content:
    • Comparison with CLT: Include a section that explicitly compares the performance of POC assays with CLT, summarizing the trade-offs in terms of sensitivity, cost, and speed.
    • Potential Innovations: Briefly outline the potential for combining POC devices with AI-driven clinical decision tools to enhance diagnostic accuracy.
  9. Minor Details:
    • Ensure the manuscript is free of typos and formatting inconsistencies. For instance, check for proper spacing in tables and figures and uniform use of abbreviations (e.g., hs-cTn, CLT).
    • Add headers or subheadings in long sections to improve navigation and clarity.
Comments on the Quality of English Language

The English language used in the manuscript is generally clear and readable, but some areas could benefit from refinement to enhance clarity and readability. Specifically:

  1. Complex Sentences:
    Many sentences are long and complex, which can make them harder to follow. Breaking these into shorter sentences will improve readability.

  2. Use of Passive Voice:
    The manuscript relies heavily on passive voice in some sections. Using active voice where appropriate can make the writing more direct and engaging.

  3. Technical Jargon:
    While the use of technical terms is necessary for a scientific audience, balancing them with simpler language in explanatory sections (e.g., introduction and discussion) could make the content more accessible to non-specialists.

  4. Grammar and Syntax:
    The grammar is correct overall, but there are occasional instances where word choices or phrasing could be streamlined for clarity. For example:

    • "The LoD is defined as the lowest concentration of analyte that the assay can reliably distinguish from a blank sample..." can be rephrased as "The LoD refers to the lowest analyte concentration reliably distinguishable from a blank sample..."
  5. Redundancy:
    There are some redundant phrases, particularly in the abstract and introduction, which could be simplified for conciseness.

Addressing these minor issues would further elevate the quality of the manuscript and ensure the research is conveyed as effectively as possible.
